# Exploring the Links Among Risky Substance Use, Problematic Internet Use, and Academic Outcomes in University Freshmen: The Role of Mediating Factors

**DOI:** 10.3390/ejihpe15060105

**Published:** 2025-06-07

**Authors:** Jessica Dagani, Chiara Buizza, Clarissa Ferrari, Giuseppe Rainieri, Alberto Ghilardi

**Affiliations:** 1Department of Clinical and Experimental Sciences, University of Brescia, Viale Europa 11, 25123 Brescia, Italy; chiara.buizza@unibs.it (C.B.); giuseppe.rainieri@unibs.it (G.R.); alberto.ghilardi@unibs.it (A.G.); 2Research and Clinical Trials Unit, Fondazione Poliambulanza Istituto Ospedaliero, Via Bissolati 57, 25124 Brescia, Italy; clarissa.ferrari@poliambulanza.it

**Keywords:** freshmen, alcohol, substance use, internet use, motivation, engagement, self-efficacy, academic dropout, academic achievement

## Abstract

**Background:** Alcohol and substance use among young people is a well-documented public health concern, and is particularly prevalent in college populations. Problematic internet use is also an emerging issue, with potential negative effects on academic achievement. University dropout remains a critical challenge, especially among freshmen, with research highlighting the role of academic engagement factors such as motivation, self-efficacy, and university connectedness in students’ academic trajectories. **Methods:** This study explored the relationships among risky substance use, problematic internet use, academic engagement factors, and academic outcomes, identifying potential mediators. Freshmen from an Italian university were invited to complete an online survey assessing these variables. The study defined two academic outcomes: (i) academic performance (Grade Point Average, GPA) and (ii) dropout intentions. Spearman’s rho coefficients and multiple linear regression models examined the associations among risky substance/internet use, academic engagement factors, and academic outcomes. Mediation analyses assessed whether academic engagement variables mediated the relationship between risky substance/internet use and academic outcomes. **Results:** The results showed that only problematic internet use was significantly associated with GPA, with self-efficacy and lack of motivation fully mediating this relationship. Regarding dropout intentions, problematic internet use and the risky use of alcohol, cannabis, and sedatives were directly and positively associated with dropout intentions. Several motivation subscales, self-efficacy, and university connectedness mediated these relationships. **Conclusions:** These findings highlight the role of academic engagement factors in mitigating the impact of risky behaviors on students’ academic trajectories, emphasizing the need for targeted prevention and intervention strategies.

## 1. Introduction

Alcohol and substance use among young people is a significant and well-documented public health concern, and is particularly prevalent in college students ([5]; [54]). In Italy, 2020 data indicate that approximately 850,000 young adults (aged 18–25) engage in risky alcohol consumption ([23]), while a 2023 national survey reported the past-year use of psychoactive substances (mainly cannabis) to be 31.8% in men and 22.5% in women in this age group ([24]). In addition to substance use, young people (especially college students) are also at an increased risk of problematic mobile phone and internet use ([33]; [55]). Recent meta-analyses and systematic reviews have demonstrated that college students exhibiting mobile phone or internet addiction are more likely to experience heightened levels of anxiety and depression ([31]; [37]). These behaviors have also been linked to poor academic outcomes ([25]; [28]; [38]). In particular, excessive internet use has been linked to lower academic self-efficacy and reduced motivation, which can negatively impact academic achievement ([14]; [49]).

School dropout is another critical issue among young adults. In higher education, this is particularly true for freshmen, as the first year is a crucial period in determining university persistence or attrition ([40]). Dropout is a multifaceted phenomenon influenced by various academic engagement factors, including motivation, self-efficacy, and a sense of connectedness with the university ([9]; [10]; [16]; [18]; [35]; [45]).

Given the well-established effects of alcohol and substance use on cognitive functioning ([19]; [56]), recent studies have explored whether these behaviors contribute to academic underperformance and dropout risk. Several studies support this link, showing that substance use is associated with lower academic achievement and an increased likelihood of dropping out ([21]; [22]; [50]; [52]). However, the evidence is not entirely consistent. For instance, [3] ([3]) found no significant differences in alcohol use between students who dropped out and those who persisted in their studies. More broadly, recent findings confirm that alcohol and substance use, particularly when frequent or severe, is negatively associated with academic performance ([1]; [8]; [27]; [29]; [34]; [36]; [39]; [43]). Yet, some discrepancies remain; for example, [44] ([44]) reported inconsistent associations between alcohol use and Grade Point Average (GPA).

A similar pattern has emerged regarding problematic internet and smartphone use. These behaviors have been linked to reduced academic self-efficacy and intrinsic motivation ([14]; [49]), as well as to poorer academic outcomes overall ([2]; [30]). Nonetheless, research in this area is still relatively limited and warrants further investigation. Importantly, most existing studies have examined substance and internet misuse in isolation, without considering their potential combined impact on students’ academic engagement. Moreover, little is known about the psychological mechanisms underlying these associations, especially in early university stages.

The effects of substance and internet misuse extend beyond cognitive impairments, as some studies suggest that these behaviors may also impact motivation and self-efficacy ([4]; [26]; [42]). However, the extent to which academic engagement factors, such as self-efficacy, motivation, and university connectedness, mediate the relationship between substance/internet misuse and academic performance has not been thoroughly explored. Addressing this gap is crucial, as identifying potential targets may inform early intervention strategies aimed at enhancing academic success and reducing dropout rates among university freshmen.

The present study aimed to address this gap by examining the interrelations among risky substance use and problematic internet use, academic engagement factors, and academic achievement, with a particular focus on identifying potential mediators. Based on the available literature, we hypothesized the following:
**H1.** *We hypothesized an association among academic engagement factors and academic achievement.*
**H2.** *We hypothesized an association among risky substance use, problematic internet use, and academic achievement.*
**H3.** *We hypothesized that academic engagement factors would mediate the association among risky substance/internet use and academic achievement.*

The findings of this study may contribute to the development of targeted interventions designed to support freshmen in achieving academic success and reducing their likelihood of dropping out.

## 2. Materials and Methods

### 2.1. Study Design

This cross-sectional study was conducted in June 2022 and involved freshmen from a university in Northern Italy. In collaboration with the University Secretariat, all enrolled freshmen were invited via email to participate in a multidimensional online survey created with LimeSurvey (https://www.limesurvey.org, accessed on 3 May 2022), ensuring anonymous data collection. The email included a link to the survey and an informative document detailing the study. Participation was voluntary, and responses were anonymous. Informed consent was obtained online before survey access. The study adhered to the World Medical Association’s Declaration of Helsinki for research involving human subjects and received ethical approval from the Board of Directors of the University of Brescia (provision no. 330, 22 November 2021).

### 2.2. Participants

Out of the 3,756 freshmen invited to participate in the study, 1618 (43.1%) accessed the survey, and 721 (19.2%) completed it. Most participants were female (63.4%) and Italian (95%). The mean age of the sample was 20.83 years (SD = 3.83). Academic characteristics are reported in Table 1.

### 2.3. Instruments

The online survey assessed socio-demographic characteristics, academic careers, and physical and mental well-being, including the following standardized instruments:World Health Organization’s Alcohol, Smoking, and Substance Involvement Screening Test v3.0 (ASSIST): this is a self-report questionnaire ([7]) designed to detect and manage substance use. It includes eight questions covering ten substances: tobacco, alcohol, cannabis, cocaine, amphetamine-type stimulants, inhalants, sedatives, hallucinogens, opioids, and “other drugs”. An example item is “In your life, which of the following substances have you ever used?”. Each substance receives a score from 0 to 36, categorized as low, moderate, or high-risk use (with higher scores corresponding to higher risk). In this study, tobacco was excluded from the analyzed substances, primarily due to its distinct social context and less acute potential for immediate academic impairment compared to alcohol and illicit substances. In our sample, 17.9% of participants exhibited a moderate-to-high-risk pattern of use for at least one substance. Specifically, 11.5% reported moderate-to-high-risk alcohol use (cut-off score = 10), 7.1% reported moderate-to-high-risk cannabis use, and 3.2% reported a moderate-to-high-risk sedative use (cut-off score = 3 for both cannabis and sedatives). For all other substances, the proportion of moderate-to-high-risk users was below 0.5%.Internet Abusive Use Questionnaire (IAUQ, [12]): this is a 12-item scale rated on a five-point Likert scale from 0 (“Totally disagree”) to 4 (“Totally agree”), assessing problematic internet use. An example item is “You lose sleep in order to stay online”. The total score ranges from 0 to 48, with the established cutoff of 24 or higher indicating problematic internet use. In this sample, the overall Cronbach’s alpha value was 0.91, and 10.7% of students scored above this cutoff.An adapted form of the Academic Motivation Scale (AMS, [10]): this was developed based on the Self-Determination Theory from [53] ([53]), it assesses motivation through five subscales. Each subscale is composed of four items rated on an 11-point Likert scale ranging from 0 (“Not at all true”) to 10 (“Completely true”). The five subscales are (i) Amotivation (i.e., lack of motivation, example item: “I can’t see why I go to school and, frankly, I couldn’t care less”), (ii) External Regulation (i.e., motivation driven by external rewards, punishments, or demands, example item: “Because someone else expects me to”), (iii) Introjected Regulation (i.e., motivation driven by internal pressure like guilt or shame, example item: “To prove to myself that I am capable of completing my high-school degree”), (iv) Identified Regulation (i.e., motivation based on personal values and conscious choice, example item: “Because eventually it will enable me to enter the job market in a field that I like”), and (v) Intrinsic Regulation (i.e., motivation arising from inherent interest, enjoyment, and satisfaction in the academic experience, example item: “Because my studies allow me to continue to learn about many things that interest me”). The total score ranges from 0 to 40, with a higher score indicating a greater adherence to the construct represented by each subscale. In our sample, the overall Cronbach’s alpha values for the five subscales ranged from 0.78 to 0.94.Perceived School Self-Efficacy Scale (SASP, [10]; [41]): this measures students’ perceptions of their ability to regulate and focus on their studies. It includes nine items rated on a five-point Likert scale ranging from 1 (“Not capable at all”) to 5 (“Fully capable”). An example item is “Focus on your studies without getting distracted”. The total score ranges from 9 to 45, with a higher score indicating a greater perceived level of self-efficacy. In this sample, the overall Cronbach’s alpha value was 0.88.University Connectedness Scale (UCS, [48]): this measures students’ perceived support and sense of belonging within their university. It consists of 18 items rated on a seven-point Likert scale, ranging from 1 (“Not at all”) to 7 (“All the time”). An example item is “Class sizes are so large that I feel like a number”. The total score ranges from 18 to 126, and higher scores indicate a stronger sense of connectedness and support. In our sample, the overall Cronbach’s alpha value was confirmed at 0.88.Freshmen’s dropout intentions were assessed using a composite score adapted from [20] ([20]) for the Italian university context ([10]). Students were asked how often they (1) Think about dropping out of college and pursuing something else; (2) Feel insecure about continuing their college studies year after year; (3) Consider the idea of discontinuing their college education; and (4) Intend to drop out of college. Each item was rated on a five-point Likert scale ranging from 1 (“Never”) to 5 (“Always”), with higher scores indicating greater dropout intentions. The dropout intention score was calculated as the mean of these four items ([15]).

Further details on the means and standard deviations of all assessment scale scores and academic outcomes are provided in Table 2.

### 2.4. Outcome Definition

This study aimed to evaluate academic engagement factors that could mediate the impact of substance and internet risky use on students’ academic trajectories. The following primary outcome measures were defined:
Academic performance, measured as GPA (continuous variable).Dropout intentions, a continuous score ranging from 1 to 5, where higher values indicate greater dropout intentions.

### 2.5. Statistical Analysis

The statistical analyses were performed by R-statistical software (version 4.4.1, 2024 The R Foundation for Statistical Computing, Vienna, Austria) and the software Jamovi (the jamovi project version 2.4, 2025, https://www.jamovi.org, accessed on 13 March 2025). Descriptive statistics were computed for socio-demographic and academic variables, as well as the questionnaires’ scores, reporting percentage distributions for categorical variables and means with standard deviations (SD) for continuous variables. Based on the observed prevalence of risky substance use and problematic internet use (as detailed in the Instruments section), the following variables were selected as direct predictors of the two outcomes: risky alcohol use, risky cannabis use, risky sedative use, and problematic internet use. Spearman’s rho coefficients and multiple linear regression models assessed the correlations between continuous outcomes, substance/internet risky use, and academic engagement variables. A stepwise model selection procedure (carried out by the stepAIC function of the R-package MASS) was used to identify the best independent predictors for each outcome. Mediation analyses were conducted to examine whether academic engagement variables mediated the relationship between risky substance use and problematic internet use with academic outcomes, following [6]’s ([6]) framework (Appendix A). The process of assessing mediation follows three steps:Step 1: Assess the association between the independent variable (i.e., risky substance use and problematic internet use) and the dependent variable (i.e., GPA and dropout intentions). This initial assessment establishes the *c*’ path, representing the direct effect of the independent variable (X) on the dependent variable (Y) before including the mediator (M, i.e., academic engagement factors).Step 2: Assess the association between X and M. This step determines the strength and direction of the *a* path, verifying that X has an effect on the proposed M.Step 3: Assess the association between M and Y while simultaneously controlling for X. This step evaluates the *b* path, confirming whether M significantly influences Y beyond the effect of X.

Based on the results of these assessments, particularly the effect of X on Y when controlling for M, the type of mediation can be determined. Full mediation occurs when, after controlling for M, X is no longer significantly associated with Y, indicating that the entire effect of X on Y is transmitted through M. Partial mediation occurs when the association between X and Y is significantly reduced but still statistically significant after controlling for M. This suggests that M accounts for part, but not all, of the relationship, leaving a residual direct effect of X on Y.

In our applied mediation analyses, we assessed whether academic engagement variables (potential mediators M) mediated the relationship between risky substance use/problematic internet use (X variables) and academic outcomes (Y variables). Each mediator—self-efficacy (SASP total score), motivation (AMS subscale scores), and university connectedness (UCS total score)—was tested in separate models to identify specific intervention targets aimed at improving academic success and reducing dropout intentions.

The associations of steps 1–3 were assessed by linear models (both univariable and multivariable linear models, with beta regression coefficients as effect sizes of the relations between X, M and Y). The level of statistical significance was set at *p* = 0.05.

## 3. Results

### 3.1. Correlation Analysis Among Primary Outcomes, Substance/Internet Risky Use, and Academic Engagement Variables

Regarding the first outcome, a significant positive correlation was found between problematic internet use and academic performance, as measured by GPA (*p* < 0.001). Additionally, problematic internet use was correlated with UCS, SASP, and all five AMS subscales, which were, in turn, correlated with GPA (see correlation matrix in Appendix A). Interestingly, no significant correlation was found between risky substance use and academic performance. These findings suggest that academic engagement variables (UCS, SASP, and AMS subscales) may act as potential mediators in the direct relationship between problematic internet use and academic performance.

Regarding the second outcome (dropout intentions), it was positively correlated (*p* < 0.001) with both problematic internet use and risky substance use, although Spearman’s rho coefficients were relatively low for alcohol, cannabis, and sedatives. Moreover, risky substance use was correlated with UCS, SASP and all five AMS subscales, which were, in turn, correlated with dropout intentions (see correlation matrix in Appendix A). These correlation analyses further suggest that academic engagement variables may mediate the relationship between substance/internet risky use and dropout intentions.

### 3.2. Mediation Models for Academic Performance

Univariable linear models were applied to assess the direct effect (Step 1 of the mediation model) of alcohol/substance risky use and problematic internet use on academic performance (GPA). Only problematic internet use was significantly associated with GPA, with a beta regression coefficient equal to −0.040 (*p* < 0.001). Additionally, the association of problematic internet use with the seven potential mediators (i.e., academic engagement variables) was examined using seven separate regression models (Step 2 of the mediation model). Significant negative correlations were found for UCS, SASP, and Identified regulation and Intrinsic regulation AMS subscales (*p* < 0.001 for all). Significant positive correlations were found for Amotivation, External regulation and Introjected regulation AMS subscales (*p* < 0.001 for all). Multivariable regression models (Step 3 of the mediation model) identified SASP and the Amotivation AMS subscale as full mediators of the relationship between problematic internet use and GPA, as the direct effect was no longer significant after adjusting for these mediators. The remaining five academic engagement variables acted as mediators, as the direct effect remained significant but was reduced in magnitude (Figure 1).

### 3.3. Mediation Models for Dropout Intentions

Similar mediation models were performed for the second outcome, namely dropout intentions. In addition to problematic internet use (Figure 2A), risky alcohol, cannabis, and sedative use was found to be directly and positively correlated to dropout intentions (Figure 2B–D). Regarding the relationships between problematic internet use and dropout intentions, and between risky alcohol use and dropout intentions, all seven mediators acted as partial mediators (Figure 2A,B). A different pattern emerged for the direct positive relationship between risky cannabis use and dropout intentions (Figure 2C): only the UCS, Amotivation AMS subscale, and External regulation AMS subscale acted as full mediators, while the remaining four variables did not satisfy the Step 2 condition of the mediation model. Notably, the UCS score exhibited the strongest mediating effect, significantly reducing the direct association between risky cannabis use and dropout intentions (beta of direct effect: 0.028, *p* < 0.05; beta after controlling for UCS: 0.013, *p* > 0.05). Finally, regarding the relationship between risky sedative use and dropout intentions (Figure 2D), five academic engagement variables acted as partial mediators, whereas the Identified regulation and Intrinsic regulation AMS subscales did not satisfy the mediation model criteria. Among the mediators, the Amotivation AMS subscale had the strongest effect in reducing the direct association between risky sedative use and dropout intentions (beta of direct effect: 0.055, *p* < 0.001; beta after controlling for Amotivation AMS subscale: 0.035, *p* = 0.007).

## 4. Discussion

This study aimed to examine the impact of risky substance and internet use on two academic outcomes: academic performance (measured by GPA) and dropout intentions. Additionally, we explored the mediating role of academic engagement factors in a large sample of Italian freshmen.

In line with our first hypothesis (H1), our results confirmed the association between academic engagement factors and academic achievement, highlighting the importance of university connectedness, motivation, and self-efficacy in fostering academic success and student retention ([9]; [17]; [47]). Regarding our second hypothesis (H2), we found that not all anticipated associations between risky substance/internet use and academic outcomes were significant. Specifically, both risky substance use (alcohol, cannabis, and sedatives) and problematic internet use were positively associated with dropout intentions, while GPA was negatively associated with problematic internet use but not with risky substance use. These findings are partly consistent with prior research. [50] ([50]) found that substance use was significantly associated with academic dropout among college students. However, the associations between substance use and academic performance have been more mixed: while [44] ([44]) reported no significant relationship between alcohol use and academic performance, [29] ([29]) observed negative associations between both alcohol and cannabis use and GPA. Regarding problematic internet use, an Iranian study found that internet addiction was inversely related to academic performance, consistent with our findings ([25]).

These findings should also be interpreted within the context of the present study. As freshmen, participants had completed a limited number of exams, meaning that their GPA may not fully reflect their long-term academic performance. Nevertheless, the association between problematic internet use and a lower GPA likely reflects difficulties in time management, concentration, and sleep quality, as suggested by research on social media use and academic performance ([25]; [13]). Additionally, the low prevalence of risky cannabis and sedative use may have limited statistical power to detect significant associations. Future research with more representative samples could clarify these effects.

Mediation models were used to assess the role of academic engagement factors in the relationship between problematic internet use and academic performance, as hypothesized in H3. Self-efficacy and lack of motivation fully mediated the negative impact of problematic internet use on GPA, indicating that this behavior influenced academic outcomes primarily through its effects on these psychological variables. Additionally, university connectedness and the other motivation subscales functioned as partial mediators. A strong sense of belonging to the university and motivation based on personal values or driven by interest and enjoyment appeared to buffer, though not eliminate, the negative impact of problematic internet use on academic performance. Conversely, motivation based on the expectation of external rewards or punishments, or driven by feelings such as guilt or shame, appeared to intensify this impact.

These findings align with recent studies showing that problematic internet use is associated with lower academic self-efficacy ([14]; [49]). However, the role of different forms of motivation in this association remains unclear in the literature, with mixed evidence. For instance, [46] ([46]) found that external motivation moderated the link between social media addiction and low personal accomplishment, such that students with low external motivation showed a stronger negative association. In contrast, [32] ([32]) reported that external motivation hindered academic performance among students with high intrinsic motivation but was beneficial for those with low intrinsic motivation. A recent review ([51]) concluded that both intrinsic and identified motivation are positively associated with academic achievement, while findings for external motivation are inconsistent, ranging from weak negative associations to no clear relationship. These inconsistencies suggest that the motivational profile of students may interact in complex ways with problematic internet use and academic outcomes, highlighting the need for further investigation.

For dropout intentions, multiple mediation models were tested. In the case of internet use and alcohol use, all mediators partially reduced the direct association, with university connectedness and self-efficacy emerging as key protective factors, while lack of motivation contributed to the negative impact of internet and alcohol use on dropout intentions.

For risky cannabis use, university connectedness, lack of motivation, and external motivation fully mediated its impact on dropout intentions, suggesting that a strong sense of belonging mitigates negative effects, while amotivation or externally controlled motivation exacerbates them. These results are consistent with findings from previous research on college students ([10]; [16]).

Finally, for sedative use, university connectedness, self-efficacy, lack of motivation, external and introjected motivation acted as partial mediators. Lack of motivation was the strongest mediator, indicating that sedative use may increase dropout intentions by fostering academic disengagement. The role of external and introjected motivation suggests that sedative use may also be linked to more controlled forms of motivation, potentially jeopardizing academic persistence.

Previous research has shown that high school students who have never used substances (including cigarettes, alcohol, and illicit drugs) report higher levels of academic self-efficacy and emotional engagement compared to their peers who have used substances in the past year ([11]). This pattern may carry significant implications for academic persistence. Moreover, both alcohol and substance use have been associated with an increased risk of school dropout in this population ([21]; [22]), suggesting that these behaviors may undermine students’ long-term academic trajectories.

Our findings extend this evidence to the university context, showing that similar patterns hold among college students. This continuity of potential risk factors across educational transitions is supported by previous studies linking substance use to an increased likelihood of academic dropout in university populations ([52]). Moreover, problematic internet use has been associated with lower academic self-efficacy and intrinsic motivation among college students ([14]; [49]), both of which are well-established predictors of dropout ([10]; [16]; [35]).

This underscores the crucial role of motivation, self-efficacy, and university connectedness in counteracting these negative effects, emphasizing the need for targeted interventions to support at-risk students and foster both academic engagement and psychological resources.

Despite its strengths, this study has some limitations. While the large sample size supports robust findings, its generalizability is limited, as participants were drawn from a single university. Additionally, voluntary participation may have introduced self-selection bias. The reliance on self-report measures, while allowing broad data collection, may be subject to biases affecting validity. Finally, the low prevalence of risky substance use, especially cannabis and sedatives, may have reduced our ability to detect certain associations or mediation effects that larger samples with higher substance use rates might reveal.

## 5. Conclusions

Our findings underscore the critical role of self-efficacy, university connectedness, and motivation, particularly the detrimental impact of low or externally regulated motivation in mediating the academic consequences of risky substance and internet misuse. These results suggest clear, actionable strategies for universities committed to enhancing student success and retention. To strengthen self-efficacy, universities should implement practical courses or workshops focused on essential academic skills such as time management, study strategies, and stress management. Addressing motivational challenges is equally crucial, with robust tutoring and mentoring programs capable of sustaining motivation and supporting students in overcoming academic difficulties. Furthermore, fostering an inclusive academic environment is paramount; this is achievable by promoting extracurricular activities, facilitating study groups, offering accessible counseling services, and organizing engaging social events that significantly enhance students’ sense of belonging. Finally, given the observed impact of problematic internet and substance use, universities must also prioritize educational campaigns to raise awareness of their potential negative effects on academic performance and well-being. These evidence-informed interventions, derived from our findings, can contribute to cultivating a more supportive and conducive learning environment, ultimately promoting student persistence and academic success from the crucial freshman year onward.

## Figures and Tables

**Figure 1 ejihpe-15-00105-f001:**
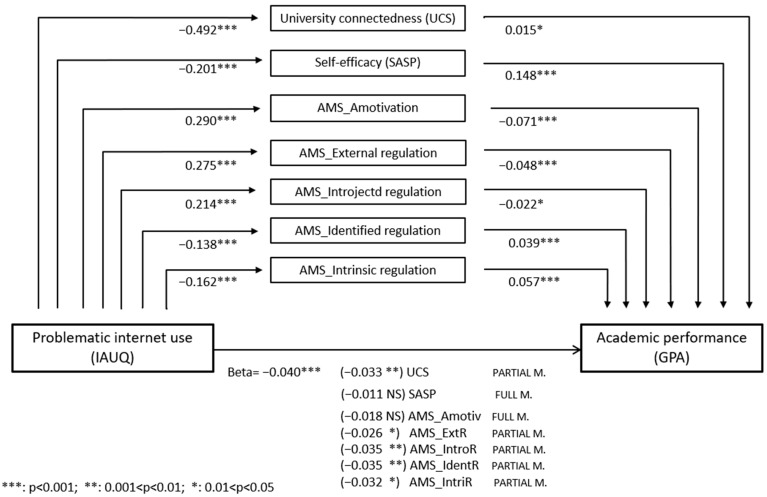
Mediation models—academic performance. Values represent the beta regression coefficients. The horizontal arrow indicates the direct effect of the exposure (Problematic Internet Use) on the outcome (Academic Performance). The central rectangles represent the mediators: seven different mediation models were performed, each considering one mediator at a time. The total effect can be computed by summing the direct effect (beta of path *c*’ = −0.04) and indirect effect (beta of path *a* × beta of path *b*; total effect = *c*’ + *a* × *b*, see Appendix A). The significance levels of the relationships for steps 1–3 of the mediation models are indicated by asterisks (*) or “NS” (Not Significant).

**Figure 2 ejihpe-15-00105-f002:**
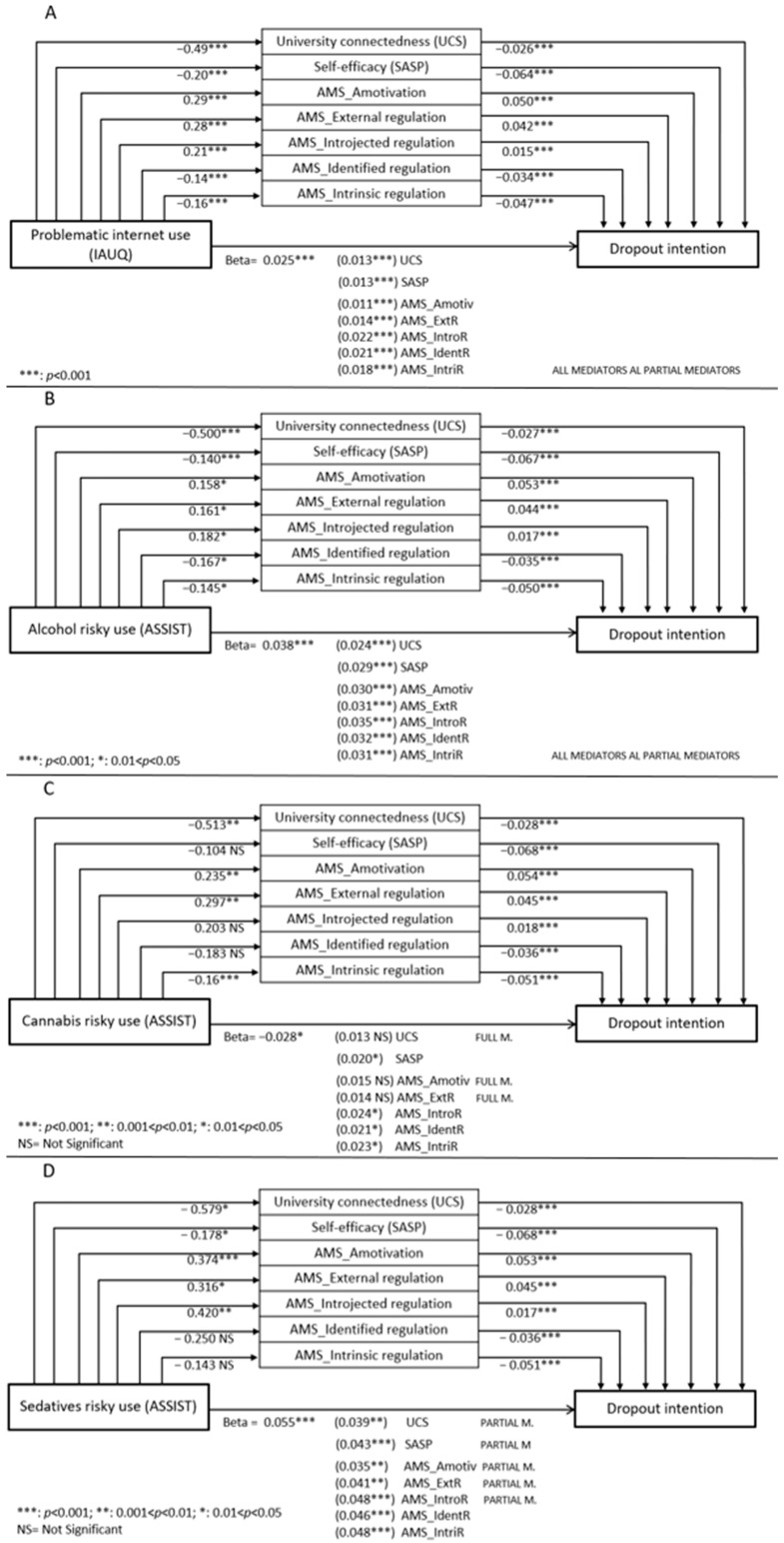
Mediation models—dropout intentions. Values represent the beta regression coefficients. The horizontal arrow indicates the direct effect of the exposures (in turn: (**A**) Problematic use of internet, (**B**) Alcohol risky use, (**C**) Cannabis risky use, (**D**) Sedative risky use) on the outcome (Dropout Intentions). The central rectangles represent the mediators: seven different mediation models were performed, each considering one mediator at a time. The total effect can be computed by summing the direct effect (beta of path *c*’) and indirect effect (beta of path *a* × beta of path *b*; total effect = *c*’ + *a* × *b*, see Appendix A). The significance levels of the relationships for steps 1–3 of the mediation models are indicated by asterisks (*) or “NS” (Not Significant).

**Table 1 ejihpe-15-00105-t001:** Academic characteristics of the sample.

Variable	N	%
Field of study		
Medicine	302	41.9
Engineering	157	21.8
Economics	180	25
Law	46	6.4
Pharmacy	23	3.2
Agricultural	13	1.8
Employment status		
Student	473	65.6
Working student	248	34.4
	**Mean**	**SD**
Grade Point Average ^1^	24.80	2.80
Percentage of attended lessons	78.67	27.85
Mean number of hours spent studying per day	3.90	2.03

Boldface indicates rows specifying the type of data reported in each column. ^1^ In the Italian university system, the passing grades for each exam or learning activity can range from 18 to 30.

**Table 2 ejihpe-15-00105-t002:** Questionnaires’ score.

Questionnaire	Mean	SD	Scale Range
ASSIST alcohol score	4.71	5.04	0–36
ASSIST cannabis score	0.84	3.33	0–36
ASSIST sedative score	0.43	2.65	0–36
IAUQ total score	12.43	9.42	0–48
AMS—subscales’ scores			0–40
Amotivation	6.21	7.67	
External regulation	5.99	5.69	
Introjected regulation	20.87	10.49	
Identified regulation	31.20	9.60	
Intrinsic regulation	30.70	8.21	
SASP total score	28.75	6.01	9–45
UCS total score	82.93	17.07	18–126
Dropout Intentions	2.18	1.01	1–5

IAUQ: Internet Abusive Use Questionnaire; UCS: University Connectedness Scale; SASP: Perceived School Self-Efficacy Scale; AMS: Academic Motivation Scale; ASSIST: World Health Organization’s Alcohol, Smoking, and Substance Involvement Screening.

## Data Availability

The data supporting the findings of this study are not publicly available due to the inclusion of sensitive or confidential information related to students. Access to the data may be granted upon reasonable request to the corresponding author, subject to ethical and legal constraints.

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
