# Peer review of "Exploring the Links Among Risky Substance Use, Problematic Internet Use, and Academic Outcomes in University Freshmen: The Role of Mediating Factors"

_ejihpe, 2025, doi:10.3390/ejihpe15060105_

Round 1
Reviewer 1 Report
Comments and Suggestions for Authors
The introduction is adequate to frame the research problem. However, it would be advisable to update some of the sources, as there are references to studies that are more than 10-20 years old. Bearing in mind that the topic addressed in the manuscript is highly topical, it should start from a theoretical and empirical basis that accurately reflects the state of the art, as up to date as possible.
The authors should explicitly present the research hypotheses, clearly formulated and operational, after the objective and before the method section. In addition, they can be numbered H1, H2, ... for easy reference later in the discussion section.
Why was tobacco excluded in the analysis of substance use?
In the description of instruments the authors could include a sample item for each of the subscales.
The description of the participants should be moved as part of the Method section (not as results).
Figure 1 should be omitted, a full description in the data analysis section is sufficient. In this section, the detailed description of all statistical analyses applied should be expanded, as well as the corresponding mentions of effect sizes. The reference to the statistical software used with its corresponding citation would also be missing.
The presentation of results is confusing and limited. The use of tables is recommended. Report the indirect effects in the mediations with their sizes and 95% CI. Indicate adjusted R² for each regression model. It would be advisable to include effect sizes in the results to improve the transparency, interpretation and applied usefulness of the study.
Include a Conclusions section, emphasizing the practical implications of the findings.
Author Response
Comment 1: The introduction is adequate to frame the research problem. However, it would be advisable to update some of the sources, as there are references to studies that are more than 10-20 years old. Bearing in mind that the topic addressed in the manuscript is highly topical, it should start from a theoretical and empirical basis that accurately reflects the state of the art, as up to date as possible.
Response 1: We thank the reviewer for this valuable feedback. We agree that updating the sources to reflect the current state of the art is important for such a topical subject. We've now updated the introduction by adding numerous recent and relevant international studies to ensure a strong theoretical and empirical basis (Page 2 and Refs. from the new list of references n. 9-13, 16, 18, 23-25, 29-31, 36).
Comment 2: The authors should explicitly present the research hypotheses, clearly formulated and operational, after the objective and before the method section. In addition, they can be numbered H1, H2, ... for easy reference later in the discussion section.
Response 2: We thank the reviewer for this helpful comment. We agree that clearly formulated and explicitly presented hypotheses significantly enhance the manuscript's structure and readability. We have revised the manuscript to explicitly present our research hypotheses (H1, H2, and H3) in the Introduction section (line 87-94). Each hypothesis is now clearly formulated and numbered for easy reference. As suggested by the reviewer, we have also ensured that these hypotheses are directly referenced and discussed in the relevant sections of the Discussion, allowing for clear interpretation of our findings in relation to our initial predictions (line 323,326,348). This change significantly improves the clarity and logical flow of our manuscript, and we appreciate the reviewer bringing this to our attention.
Comment 3: Why was tobacco excluded in the analysis of substance use?
Response 3: We thank the reviewer for this comment. While acknowledging tobacco's significant public health implications, its exclusion was based on its distinct social context and acute potential for academic impairment compared to alcohol and illicit substances. Our primary focus was on forms of substance use with a more direct and immediate impact on cognitive functions, concentration, and daily academic behaviors. Alcohol and illicit substances typically exert more acute effects that can directly interfere with study habits and academic performance. The social patterns and immediate behavioral consequences of problematic alcohol and illicit substance use within a university setting also differ considerably from those typically associated with tobacco. Therefore, to maintain a cohesive construct of 'risky substance use' relevant to acute academic outcomes and engagement, tobacco was not included in this particular analysis. We specified it in the Instruments section (line 130-132).
Comment 4: In the description of instruments, the authors could include a sample item for each of the subscales.
Response 4: Following the reviewer’s recommendation, we have integrated an example item for each assessment tool (and subscale) to provide further context for the reader (line 127,141,153-159,166,172).
Comment 5: The description of the participants should be moved as part of the Method section (not as results).
Response 5: We appreciate the reviewer's valuable suggestion. We agree that the description of the participants and the associated descriptive statistics belong in the Methods section for better clarity and logical flow. We've now moved the sample sociodemographic and academic characteristics (and Table 1, as well) to a dedicated "Participants" subsection within the Methods section (line 111-119). Furthermore, the descriptive statistics for all questionnaire scores have been integrated into the "Instruments" subsection of the Methods, where each measure is described. The comprehensive table summarizing these variables (Table 2), was also moved in the "Instruments" subsection (line 185-190). This reorganization ensures that all relevant data characteristics are presented before the analytical results.
Comment 6: Figure 1 should be omitted, a full description in the data analysis section is sufficient.
Response 6: We thank the reviewer for this valuable feedback. As per their suggestion, we have removed (and slightly modified) the figure from the main text and incorporated a detailed description of the model into the 'Statistical Analysis' section (line 215-236). We moved the figure in the supplementary materials (Figure S1). We believe this improves the clarity and flow of the manuscript.
Comment 7: In this section, the detailed description of all statistical analyses applied should be expanded, as well as the corresponding mentions of effect sizes. The reference to the statistical software used with its corresponding citation would also be missing.
Response 7: Thanks for this useful suggestion that improves the readability and reproducibility of our analyses and results. We've now extended the description of the applied statistical methods in the 'Statistical Analysis' section, specifying the statistical models (and corresponding effect size) used for assessing the mediation effects (line 199-242)
Comment 8: The presentation of results is confusing and limited. The use of tables is recommended. Report the indirect effects in the mediations with their sizes and 95% CI. Indicate adjusted R² for each regression model. It would be advisable to include effect sizes in the results to improve the transparency, interpretation and applied usefulness of the study.
Response 8:
We sincerely thank the reviewer for this helpful and detailed comment. We appreciate the suggestion to improve the presentation of the results, and we agree that clarity and completeness are essential. While some of the other reviewers considered the current presentation to be clear, we understand that the inclusion of tables could enhance readability. However, we were concerned that additional tables might risk making the presentation more complex or redundant.
Regarding the mediation analyses, we followed the conventional approach of reporting beta regression coefficients for each path (i.e., direct effect c’, paths a and b), as recommended in classic mediation references (e.g., Baron & Kenny, 1986; MacKinnon, 2008). As noted, based on the assumption described by Baron and Kenny, mediation effects are typically not assessed using R². However, the strength of the relationships among variables is indicated in the correlation matrix reported in supplementary materials.
Regarding the comment on reporting indirect effects, we would like to point out that Figures 1 and 2 present multiple mediation models. Providing a table with the direct, indirect, and total effects for each model (i.e., five multiple mediation models, each comprising seven separate mediation analyses= 35 tables) would be extremely complex and would compromise the readability of the results. In our view, such a substantial addition of data would not offer additional informative value to justify its inclusion. However, while the manuscript already reports the coefficients for the relevant paths, we have now added clarifications in the footnotes of Figures 1 and 2 on how the total and indirect effects can be computed based on these parameters (line 284-285, 314-316).
Baron, R. M.; Kenny, D. A. (1986). The Moderator-Mediator Variable Distinction in Social Psychological Research: Conceptual, Strategic, and Statistical Considerations. Journal of Personality and Social Psychology. 51 (6): 1173–1182. doi:10.1037/0022-3514.51.6.1173
MacKinnon, D. P. (2008). Introduction to Statistical Mediation Analysis. New York
Comment 9: Include a Conclusions section, emphasizing the practical implications of the findings.
Response 9: We thank the reviewer for the helpful suggestion to create a distinct Conclusions section. We agree that this enhances the manuscript's structure and readability. We have now separated the concluding remarks from the main Discussion, establishing a dedicated "Conclusions" section at the end of the manuscript and emphasizing the practical implications of the findings (line 413-431).
Reviewer 2 Report
Comments and Suggestions for Authors
The present study examined the link among risky substance use, problematic internet use and academic outcomes (GPA and dropouts) in an Italian University freshmen, focusing on identifying potential mediators. Authors report that only problematic internet use was significantly associated with GPA, whereas problematic internet use, risky alcohol, cannabis and sedatives use was directly/positively linked with dropout intentions. Furthermore, self-efficacy and amotivation mediated former, self-efficacy and amotivation and university connectedness mediated the latter relationship.
While the study is based on an interesting premise, it has several limitations as addressed in the study. I have few comments as listed below.
Line 19, title and several places – between or among?
Line 27, 30 and several places – motivation or amotivation?
Line 93 – Brescia University or University of Brescia?
Line 110 – Rewrite for clarity.
Line 167 – Section 3.1 should be a part of method section.
Line 306 – Which substances?
Line 318 – Single?
Was there any effect of the employment status and the field of study on measured outcomes?
Minor grammatical, punctuation errors throughout the manuscript.
Author Response
Comment 1: Line 19, title and several places – between or among?
Response 1: We appreciate the reviewer's valuable comment regarding the consistent use of 'among' and 'between.' We've thoroughly revised the entire manuscript with particular attention to this distinction. We agree that clarity and precision are paramount in scientific writing. Rather than adopting a single term throughout, we opted to differentiate the use of 'among' and 'between' based on the specific nature of the relationship being described:
- A) We've used 'among' (e.g., in the title, abstract, and hypotheses section) when referring to complex interrelations or a network of associations involving three or more entities or categories of variables (e.g., problematic internet use, substance use, and academic engagement factors, all related to academic outcomes). We believe this term better captures the multifaceted and systemic nature of our study.
- B) We've maintained 'between' (e.g., in the results section when discussing specific correlations or mediated relationships) when referring to direct connections or comparisons between two distinct entities or two defined sets of variables.
We believe this differentiated application is more grammatically and statistically precise, ultimately contributing to greater clarity for the reader. We've implemented these changes consistently throughout the manuscript to align with this logic.
Comment 2: Line 27, 30 and several places – motivation or amotivation?
Response 2: We thank the reviewer for raising this important point regarding the terminology used. We understand the potential for ambiguity and appreciate the opportunity to clarify. When we use the term "motivation," we are referring to the broader psychological construct, which encompasses various types of motivation. Conversely, "amotivation" specifically refers to the lack of motivation. This distinction is critical as amotivation is also a specific subscale within the instrument used to measure motivation in our study. Throughout the manuscript, we have aimed to ensure this distinction is clear. Specifically, when we discuss amotivation, we are referring to the scores obtained on this particular subscale. When we refer to the broader construct of motivation, we use the general term. We have also revised the abstract and other relevant sections to explicitly highlight this differentiation, thereby minimizing any potential misinterpretation.
Comment 3: Line 93 – Brescia University or University of Brescia?
Response 3: We appreciate the reviewer highlighting the inconsistent naming convention for the university. We have carefully reviewed the manuscript and ensured that "University of Brescia" is used consistently throughout for accuracy (now line 108).
Comment 4: Line 110 – Rewrite for clarity.
Response 4: We appreciate the reviewer's request for clarification regarding Line 110. We agree that precision is important, and we've rephrased the whole description of the instrument to ensure clarity (line 145-162).
Comment 5: Line 167 – Section 3.1 should be a part of method section.
Response 5: We appreciate the reviewer's valuable suggestion. We agree that the description of the participants and the associated descriptive statistics belong in the Methods section for better clarity and logical flow. We've now moved the sample sociodemographic and academic characteristics (and Table 1, as well) to a dedicated "Participants" subsection within the Methods section (line 111-118). Furthermore, the descriptive statistics for all questionnaire scores have been integrated into the "Instruments" subsection of the Methods, where each measure is described. The comprehensive table summarizing these variables (Table 2), was also moved in the "Instruments" subsection (line 185-190). This reorganization ensures that all relevant data characteristics are presented before the analytical results.
Comment 6: Line 306 – Which substances?
Response 6: We thank the reviewer for their comment. We have now specified the substances that were considered in the cited study to enhance clarity (now line 388).
Comment 7: Line 318 – Single?
Response 7: We appreciate the reviewer's keen eye in catching the missing word in that sentence. We've added "university" to complete the sentence, making it clear that participants were drawn from a single institution (line 407). This oversight has been corrected to enhance clarity.
Comment 8: Was there any effect of the employment status and the field of study on measured outcomes?
Response 8: We thank the reviewer for the question. We did not include employment status or field of study in our mediation models, as our primary focus was on academic engagement variables. Regarding field of study, preliminary analyses showed significant differences in academic outcomes (GPA and dropout intentions) only between medical and economics students. No consistent or significant differences emerged among the other study programs, likely due to the varying and often small sizes of the subgroups. As for employment status, although we observed a difference in dropout intentions between working and non-working students (with no significant differences in GPA), the wide variability in the type of employment and number of working hours per week made it difficult to draw meaningful conclusions. For this reason, we chose not to include employment status in the main analyses.
Comment 9: Minor grammatical, punctuation errors throughout the manuscript.
Response 9: We have thoroughly reviewed the manuscript and made the necessary grammatical and punctuation corrections throughout.
Reviewer 3 Report
Comments and Suggestions for Authors
Thank you for the opportunity to review this manuscript. The topic is interesting and important. The text is well-written and well-structured; however, several key areas need improvement before publication. I have summarized my thoughts in the hope that the feedback will be helpful.
Introduction: The introduction is too long, and I recommend highlighting the novelty/significance of the research study.
Literature review: The study does not include a literature review section. Although the introduction provides a certain theoretical background, it should be extended.
Methodology: The rationale behind the approach and the specification of the objectives are logical and appropriate. The research model, methods, and techniques are functional.
Results: The results are presented clearly. The findings presented are interesting and can be considered a contribution to current knowledge.
Discussion: The results obtained should be compared with those of previously conducted international research studies.
Conclusion: There is no Conclusion section; the conclusions are part of the Discussion section. I recommend dividing the text.
References: I recommend adding resources published in 2024 and 2025.
Author Response
Comment 1: Introduction: The introduction is too long, and I recommend highlighting the novelty/significance of the research study.
Response 1: We thank the reviewer for their feedback on the introduction. We've significantly expanded the background with additional, particularly recent, references to provide a more comprehensive overview of the literature. Given these necessary additions, we weren't able to shorten the introduction. However, we have taken your suggestion to heart and have put a greater emphasis on the novelty and significance of our research. We have also made a careful effort to reduce redundant or repetitive phrasing that would unnecessarily lengthen the introduction without adding substantial content.
Comment 2: Literature review: The study does not include a literature review section. Although the introduction provides a certain theoretical background, it should be extended.
Response 2: Thanks for this valuable feedback. We agree that a more extensive theoretical background will strengthen the manuscript. We've now significantly expanded the introduction to include a more comprehensive literature review, incorporating references to recent studies (Page 2 and Refs. from the new list of references n. 9-13, 16, 18, 23-25, 29-31, 36).
Comment 3: Methodology: The rationale behind the approach and the specification of the objectives are logical and appropriate. The research model, methods, and techniques are functional. Results: The results are presented clearly. The findings presented are interesting and can be considered a contribution to current knowledge
Response 3: We are very grateful for the reviewer's positive and encouraging feedback on our Methodology and Results sections. We are particularly pleased that the rationale, objectives, research model, methods, and techniques were found to be logical and appropriate.
Comment 4: Discussion: The results obtained should be compared with those of previously conducted international research studies.
Response 4: Thanks for this important point. We've now compared our results with relevant and recent international research studies in the Discussion section, as suggested. (Refs N. 9,12, 13, 23,24, 26, 36, 31, 53,54,55)
Comment 5: Conclusion: There is no Conclusion section; the conclusions are part of the Discussion section. I recommend dividing the text.
Response 5: We thank the reviewer for the helpful suggestion to create a distinct Conclusions section. We agree that this enhances the manuscript's structure and readability. We have now separated the concluding remarks from the main Discussion, establishing a dedicated "Conclusions" section at the end of the manuscript (line 413-431)
Comment 6: References: I recommend adding resources published in 2024 and 2025.
Response 6: Thanks for this suggestion. We've added numerous recent references, particularly from 2024 and 2025, to both the Introduction and Discussion sections (Refs. N. 9-13, 16,18,23-25,29-31,36,53-55).
Round 2
Reviewer 1 Report
Comments and Suggestions for Authors
The authors have responded adequately to my recommendations.
Thanks
Reviewer 3 Report
Comments and Suggestions for Authors
I appreciate the authors' effort to implement all my suggestions and recommendations. The article meets the standards in this form, and no further changes or modifications are needed.